# Daily Human Activity Recognition Using Non-Intrusive Sensors

**DOI:** 10.3390/s21165270

**Published:** 2021-08-04

**Authors:** Raúl Gómez Ramos, Jaime Duque Domingo, Eduardo Zalama, Jaime Gómez-García-Bermejo

**Affiliations:** 1CARTIF Technological Center, 47151 Valladolid, Spain; jaiduq@cartif.es (J.D.D.); ezalama@eii.uva.es (E.Z.); jaigom@eii.uva.es (J.G.-G.-B.); 2ITAP-DISA, University of Valladolid, 47002 Valladolid, Spain

**Keywords:** HAR, neural network, LSTM, smart home, binary sensors, deep learning, CASAS

## Abstract

In recent years, Artificial Intelligence Technologies (AIT) have been developed to improve the quality of life of the elderly and their safety in the home. This work focuses on developing a system capable of recognising the most usual activities in the daily life of an elderly person in real-time to enable a specialist to monitor the habits of this person, such as taking medication or eating the correct meals of the day. To this end, a prediction model has been developed based on recurrent neural networks, specifically on bidirectional LSTM networks, to obtain in real-time the activity being carried out by the individuals in their homes, based on the information provided by a set of different sensors installed at each person’s home. The prediction model developed in this paper provides a 95.42% accuracy rate, improving the results of similar models currently in use. In order to obtain a reliable model with a high accuracy rate, a series of processing and filtering processes have been carried out on the data, such as a method based on a sliding window or a stacking and re-ordering algorithm, that are subsequently used to train the neural network, obtained from the public database CASAS.

## 1. Introduction

From the 1990s onwards, the cost associated with the manufacture of microprocessors fell so sharply that the use of computers and computer systems for home use became cost-effective [1]. The use of technology within the home improved the quality of life of its residents, allowing them to manipulate their homes remotely or improve their level of safety [2]. For most of the population, their home is a symbol of independence, social integration or an indicator of healthy living. Therefore, taking into account the situation of elderly people living alone, it is very important that their homes are equipped with the necessary services to satisfy their needs.

The capacity to analyse the activities carried out by users is one of the most studied fields in the scientific areas of machine learning or computer vision [3]. This field allows us to learn a lot of information about users, such as their psychological state or their personality. Most of the activities carried out by a given user on a daily basis can be automated or simplified as long as they can be recognised by a Human Activity Recognition (HAR) system [4]. By knowing the sequence of activities that a certain user carries out during the day, it is possible to extract important information about his/her health or to know if she/he has followed healthy habits [5].

Monitoring the activities performed by an elderly person can inform, in real-time, of such undesired behaviour as falls in the same way that equipment failures (both personal and household) can be alerted, which also leads to improved safety [6]. Being aware of such events can improve the independence of this population, thereby improving their sense of safety and comfort. Applying HAR technology can improve the quality of life of elderly people, since, by interpreting the activity they want to perform, automation can be applied to facilitate the task and save them time [7].

Current HAR techniques present several challenges that are trying to be solved [8]. One of the main problems in HAR is the fact that activities can be executed in different ways and in different sequences depending on the user. In addition, human behaviour is very difficult to model, as it can change depending on the day, user mood or any other circumstance. HAR machine vision systems have the problem of a lack of user privacy and are very sensitive to changes in lighting, shadows and occlusions [9]. The use of low-power sensor networks deployed in the house allows the collection of large amounts of data. Data analysis by deep learning methods allows activity recognition taking into account variability of the environment without compromising the privacy of the user.

The proposition in this paper is to improve the current techniques for the recognition of activities within the home in order to obtain a reliable model, with a high accuracy rate and with a functional capacity, which can act as a tool capable of improving the well-being of elderly people living alone and to facilitate the work of the team in charge of their care. The main contributions made in the present paper are the following: (i) The technology used for HAR is non-intrusive, prevailing the sense of autonomy and security of the user, avoiding the use of cameras or microphones that invade his/her privacy. (ii) The developed model is able to deal with activities of arbitrary duration. For example, the activity “Sleep” has a duration of hours, while the activity “Eat” takes only a few minutes. This aspect has been solved by using a sliding window method, which is a data pre-processing technique that makes activities independent of their duration. (iii) The system developed takes into account the time reference in which the activity takes place. For example, the activity “Morning Meds” and “Evening Meds” are identical at the execution level. However, by using the time of day in the model, this problem can be solved. (iv) The system can be used in real-time, providing an activity response with a time interval coinciding with the sampling time of the sensors, in order to detect any problem or anomalous situation as soon as possible, in contrast to other existing methods that need to know the duration of the activity in order to make the prediction. (v) The system is able to take data from past and future situations for model training due to the use of bidirectional LSTM networks. This has an impact on performance and overall results. (vi) The developed system is capable of detecting a wide range of activities (16) that can take place throughout the entire time period and that can occur in all areas of the house, in order to be able to monitor the user at any time and under any circumstances. (vii) The development has been carried out on the basis of a public dataset called CASAS. In this way, it has been possible to compare the performance of the system developed in this paper with the performance of the methods developed by other authors. As a result, it has been shown that the developed method gives better results.

The article is structured as follows: Section 2 analyses the state-of-the-art of the different techniques related to this paper. Section 3 details the techniques proposed for user activity recognition from sensor information. Section 4 analyses the experiments carried out with the proposed system and explains the results obtained. Finally, Section 5 presents the advantages and improvements of the developed system and the possible future improvements that can be made.

## 2. Overview of Related Work

Activity recognition has been a very important field of study for the last few years as it proposes an improvement in the quality of life of people. Knowing the activities that a certain person carries out as part of a daily routine has an impact on improving the knowledge of his or her state of health and physical condition. For example, in the case of a patient with dementia or mental pathologies, it is possible to analyse his/her daily routine in order to avoid undesirable consequences [10]. It is important to closely monitor the behavioural pattern of people with obesity, heart problems or diabetes to see if they are doing the exercise routine required for their treatment [11]. Having a safe and intelligent home with a wide network of sensors able to monitor activities of daily life provides occupants with an opportunity to live more safely and independently, especially those suffering from Parkinson’s or Alzheimer’s disease [12].

Depending on the environment where the activity recognition is to be performed and the type of activity to be monitored, some types of sensors will be preferred over others. For example, in a hospital, it will be necessary to recognise whether users take their medication at the correct times. In the same way, if the environment is a smart office, it is necessary to consider whether the use of resources is optimal in order to obtain the maximum benefit. Finally, taking into account the fact that the environment is a smart home, it is necessary to select a suitable sensor to reliably detect the daily habits of its occupants. In order to know the area of the house occupied by a user, the most reliable sensor to use is the Passive Infrared Sensor (PIR) [13]. The use of several sensors makes it possible to determine the path or route taken by the user in the house and, with the information from other sensors, to estimate the activity that she/he is carrying out. The use of this type of sensor is recommended because they are easy to install, are immune to pets and have a low consumption [14]. To detect activities such as leaving home, opening the refrigerator or cabinets, the simplest way is to use contact switches. Finally, the use of pressure plates to detect the use of chairs, beds or sofas is also recommended.

Although it is possible to deploy a network of sensors according to the needs of each user and the characteristics of the home, the most appropriate way to validate a new methodology is to use a public database. This allows comparison with other existing works that use the same database and makes it easier to fine-tune a new method [15]. There are a number of databases in the cloud with numerous records of the daily lives of different users. For example, in [16], a living lab at Cambridge with a living room, kitchen, bedroom and most relevant rooms is described. In this living lab, studies of a different nature are carried out: sensor-based, interview-based or direct observation studies. Sensors such as contact sensors in the refrigerator, temperature and humidity sensors or light sensors are used. On the other hand, the ARAS (Activity Recognition with Ambient Sensing) dataset has been elaborated by sampling two real houses with multiple residents for two months. Each house contains 20 binary-type ZigBee sensors [17]. In this dataset, 27 different types of activities are tagged. Another relevant dataset to consider is UvA [18]. Sensor data from three different houses were collected in this dataset over 2 months. However, these houses were only inhabited by a single resident at the time of the study and are labelled with fewer activities compared to ARAS (16 vs. 27) [19]. However, it is difficult to detect activities of short duration, so a limit of 15 min is set to obtain consistent activities. Another interesting dataset to consider is Domus [20]. This dataset is used to obtain information about the comfort and sensations of the residents inside the house from the information of air quality sensors, indoor temperature sensors or brightness sensors [21]. The perceptions to be examined are the following: global comfort, thermal comfort, lighting comfort, air quality or acoustic comfort. In terms of sensations, variables such as temperature, humidity and odour are studied for 24 different users, ranging in age from under 20 to over 60 years old. After studying the different datasets, we opted for the CASAS dataset due to the high number of sensor records it has. In addition, it has been used in several research works with similar objectives to this work, which facilitates comparisons. This dataset has different variants. Each variant presents particular circumstances, analysing different activities, with one or several resident persons and different sensors in terms of number and type [22]. It is the most advisable dataset because of its versatility and because it presents a good sensor-activity average. For example, the Milan variant has 33 sensors inside the study house where one person and his pet live, 15 different activities are recorded, and the study period is approximately 84 days. This variant has been used in this work because it is a dataset obtained for a house with one person and his/her pet, which causes noise in the sensors that can test the reliability of the model.

The objective that arises is, from a set of sensor measurements in which the sequence of activities have been labelled in a supervised way, to be able to infer a model capable of predicting new activities from new sensor measurements. There are different methods for building a model to do this. There are three types of classification methods: the threshold-based method, machine learning techniques and Deep Learning (DL) methods [23]. The threshold-based method is typically used to detect postures, movements and simple gestures, while machine learning techniques are data analysis methods whose model building is automated. For example, over the years, the Hidden Markov Models (HMM) [24] have become one of the main techniques for modelling one and two-dimensional signal prediction systems in which the information is incomplete. This type of model presents a mathematical structure based on statistical techniques whose algorithms for model parameter estimation are very easy to train. Another example is the models based on Support Vector Machines (SVM) [25], whose algorithms have a very high success rate in small sample size classification tasks. However, deep learning methods have several advantages over the methods explained above, since they achieve greater accuracy, giving rise to more accurate and reliable models such as those based on Convolutional Neural Networks (CNN). Traditional Artificial Neural Networks (ANN) methods typically have fewer internal layers than deep learning methods [26]. A higher number of layers favours the learning of large datasets. One of the keys to deep learning applications in image or video analysis is the use of networks whose architecture is based on convolutional layers [27]. This type of architecture combines convolutional layers with pooling layers. In the same way, a method to improve the results is the use of dropouts, which induce a certain amount of noise in the network that makes it behave better in the presence of data not used in the learning process. However, the models used for the analysis of sequential patterns are Long Short Term Memory (LSTM) networks. The LSTM network is a type of Recurrent Neural Network (RNN) that presents a concept of an internal memory unit, which is very beneficial when dealing with both short and long term temporal sequences [28]. A good example is the case of [29], in which the authors compare the neural network method of the LSTM type with the classic Naive Bayes method, obtaining a very relevant increase in accuracy in the former case. Another type of architecture used for HAR is the case of CNN-LSTM networks. CNN-LSTM networks are composed of a first stage of convolutional layers followed by unidirectional LSTM layers. This type of network is typically applied for image analysis due to the powerful advantage of applying successive convolutions in this field, although it can also be applicable for discrete data analysis with lower results. If a DL architecture is composed of a set of unidirectional LSTM layers together with a number of dense layers, the architecture is called an sLSTM network [30]. This type of network is typically used to reinforce information from past events due to the increased number of unidirectional LSTM layers. If the last dense layers are removed from this architecture and only the blocks of unidirectional LSTM layers are left, a model called Deep LSTM is obtained [31]. This type of architecture improves on the advantages of a conventional vanilla LSTM (vLSTM) by behaving better in the face of noise in the input data model.

There are currently several research groups that are working on different solutions to recognise the most common activities that people perform in their daily lives, both in their homes and in their exercise routines, for example. The work done by M. Ronald et al. [32] focuses on the development of a DL architecture known as “iSPLInception”. This architecture is based on an Inception-ResNet network, which is used not only for image processing but also for any system that uses complex data. The system developed by the authors improves the activity prediction results for four different datasets compared to the use of more conventional DL architectures, such as the use of CNN or vLSTM networks. The authors achieve a success rate of 95.09% for the UCI dataset, recognising a total of six activities. For the Opportunity dataset, they achieve a success rate of 88.14%, recognising a total of 17 activities. In the case of the Daphnet dataset, they achieve a success rate of 93.52%, recognising a total of two activities. Finally, for the PAMAP2 dataset, they recognise a total of 11 activities with a success rate of 89.09%. Another interesting work to take into account is the one developed by Z.Chen et al. [33], which focuses on the development of a system capable of recognising activities by means of the signals provided by a smartphone. In addition, the authors also develop a series of algorithms known as maximum full a posteriori (MFAP) to improve the model response and accuracy. To test the validity of their model, the authors make predictions on a public database known as UCI [34], improving the approaches of similar technologies. It is interesting to analyse the work done by O.Steven Eyobu et al. [35] due to their elaboration of an activity detection system by analysing a wearable sensor that collects the data provided by an inertial measurement unit (IMU). The authors use data augmentation techniques to add value to the data collected by the IMU and use a neural network-based DL model with unidirectional LSTM layers to make predictions. The system developed by the authors improves the results obtained by 52.77% if the UCI dataset is introduced without any data augmentation technique.

The work approached in this paper presents an accurate and reliable method for the detection of the activities carried out by a person living with his/her pet during their daily life. Among the contributions, it is important to highlight that a prediction model has been developed that can be used to detect activities in real-time with a very high accuracy rate. Compared to other authors who use the same database [36], who require a set of fixed records to make estimates of the activity, our model uses a sliding window, which allows real-time predictions. This is of great interest for triggering actions or recommendations to the user. In addition, the model is capable of detecting a wide range of the main activities that take place in a house on a daily basis by analysing data provided by non-intrusive sensors, without the need for cameras or microphones that invade the privacy of the subject. One of the main challenges faced by the system developed in this paper is the noise that non-intrusive binary sensors can present. Several public datasets are collected from houses that have animals. This can introduce noise into the system that can cause confusion when a critical activity is detected. Another challenge faced by the system is to bring flexibility to the model, as a specific activity can be performed in different ways.

## 3. Analysis of the System

The method proposed in this section addresses the detection of activities of a person living in a sensorised house. First, the components and tools used to achieve this objective are explained. Then, the pre-processing and filtering methods used to train the model are submitted. Finally, the architecture of the model developed for recognition is explained.

### 3.1. Components and Tools

In this section, the database chosen to extract the information used to develop the activity prediction model is explained, as well as which tools and methods have been developed to elaborate the prediction model.

#### 3.1.1. CASAS Dataset

The CASAS dataset presents a simple architecture that facilitates the deployment of sensors inside a home. This smart home project is a multidisciplinary research project at the University of Washington, which consists of the massive collection of data through sensors so that, after performing an analysis and following certain rules, it is possible to interact with a series of actuators [37]. The architecture offers a lightweight design that facilitates the implementation and development of future smart home technologies [38]. It is easy to install and ready to use without training or customisation. In the same way that [39] points out, this dataset consists of a collection of simple sensor signals within a controlled environment. These simple signals are received from motion or light sensors, for example. As explained in Section 2, the Milan variant has been chosen to build the model because there are a total of 2310 activity occurrences in it [40]. Moreover, this dataset presents some noise, since the household has a pet and the data disturbances from the sensors have to be taken into account. It is also specified that the son of the person residing in the house visits it from time to time. The main features of the Milan variant are summarised in Table 1.

Specifically, a total of 28 motion sensors, 3 door sensors (one placed in the medicine drawer) and 2 temperature sensors are distributed throughout the house, making a total of 33 sensors. When recording the measurements, the letter M is used for the motion sensors, D for the door sensors and T for the temperature sensors.

The dataset collects the events generated by the sensors in the house. That is to say, it is not a fixed sampling, but every time a sensor detects a change in its status or a measurable value, a new record is registered in the database. A record includes the following fields: the time at which the event was generated, the sensor that was affected, the new status of the sensor and the last field is reserved for recording the start or end of an activity. The format can be seen in Table 2.

With respect to the activities, only the start and end times are recorded. The rest of the time, the cells corresponding to the activity field are empty, so it is important to differentiate between whether a specific activity is being carried out or the user is simply doing nothing. To do this, if the activity field of the record being analysed is empty, it is necessary to take into account whether a ’begin’ appears in previous records (indicating that the same activity is still being carried out) or an “end” appears (indicating that the user is doing nothing).

#### 3.1.2. Neural Network—LSTM Model

Models based on neural networks built with LSTM layers are advantageous for processing temporal sequences. LSTM networks are a type of recurrent neural network (RNN) whose field of application ranges from language modelling or automatic translation to speech recognition. Because these types of networks have millions of parameters in their architecture, a large computation time is required using multi-GPU systems [41]. There are components called memory blocks and units called gates inside an LSTM cell. The memory blocks allow the temporary storage of the network states while the gates are in charge of directing the flow of information [42].

Figure 1 shows what the inside of an LSTM cell looks like, as well as the parameters that identify it. The parameters of the LSTM can be calculated as follows:(1)$$\begin{array}{c} i_{t}=\sigma \left(W_{i}x_{t}+U_{i}h_{t-1}+b_{i}\right) \end{array}$$
(2)$$\begin{array}{c} z_{t}=\tanh\left(W_{z}x_{t}+U_{z}h_{t-1}+b_{z}\right) \end{array}$$
(3)$$\begin{array}{c} f_{t}=\sigma \left(W_{f}x_{t}+U_{f}h_{t-1}+b_{f}\right) \end{array}$$
(4)$$\begin{array}{c} C_{t}=i_{t}*z_{t}+f_{t}*C_{t-1} \end{array}$$
(5)$$\begin{array}{c} o_{t}=\sigma \left(W_{o}x_{t}+U_{o}h_{t-1}+V_{o}C_{t}+b_{o}\right) \end{array}$$
(6)$$\begin{array}{c} h_{t}=o_{t}*\tanh\left(C_{t}\right) \end{array}$$

The parameters from Equations (Equation 1)–(Equation 6) are calculated iteratively from time t=1 to *T*, where $i_{t}$ corresponds to the input gate signal, $z_{t}$ is the input block information, $f_{t}$ is the forget gate signal, $C_{t}$ is the output block information, $o_{t}$ is the output gate signal and $h_{t}$ is the set of values that stores the information of the past states. The parameters $U_{i}$, $U_{o}$, $U_{f}$, $U_{z}$ and $W_{i}$, $W_{o}$, $W_{f}$, $W_{z}$ are computed during model training, while *b* is the bias and sigma and tanh are the activation functions.

If the sequence before ($t_{-1}$) and after ($t_{1}$) is available for a given time ($t_{1}$), it is possible to improve the performance of the neural network by replacing the architecture formed by conventional LSTM layers with bidirectional LSTM layers. To train this type of network, it is necessary to make passes in both directions with the dataset [43]. In this way, the model is trained taking into account sequences not only prior to the analysed instant but also after it. A very favourable advantage of bidirectional LSTM networks is the fact that a smaller buffer than conventional LSTM networks is needed to obtain similar outcomes, which results in a good response to causal systems, provided that the required output latency for the system is short [44]. This type of network is very beneficial for tasks such as emotion recognition in speech or noise modelling.

Figure 2 shows the architecture of a bidirectional LSTM network with (*T*) stages. For example, for an instant (*t*), the forward layer uses the data of instant (t−1) to generate the data of instant (t+1). However, with respect to the backward layer, the data of the instant (t+1) is used to generate the data of the instant (t−1). It is necessary to add that both processes share the same activation layer, independently of the direction of the information flow.

### 3.2. Data Processing and Filtering

The dataset selected for the development of the paper contains a very large number of records. This reason makes it necessary to process the records before feeding them into the neural network. As explained in Section 3.1.1, the dataset provides the events recorded by the sensors each time the enviroment conditions change (person arrival or a temperature change). However, the ultimate goal is to obtain the activity being performed by a given user each time a time range (*T*) is exceeded. Therefore, it is necessary to train the model with similar conditions to those sought as the final result. This leads to converting the set of events in the dataset into a dataset sampled at a fixed time (*T*).

To elaborate the dataset to train the proposed prediction model, it is necessary to transform the state of the sensors (ON–OFF for motion sensors, OPEN–CLOSE for door sensors and a numerical value for temperature sensors) into a numerical value that can be handled by the model. Therefore, for the motion and door sensors, a binary value is assigned depending on the state in which there are found at the time of sampling. With respect to the temperature sensors, it is beneficial to normalise the values between 0 and 1 to avoid using outliers for training, as all other sensor information is within this range.

In addition to the status of the sensors, it is also important to consider the time of day at which the record was taken, as this can be a key factor in differentiating activities. The importance of the time of day is due to the fact that the user has a very similar pattern of activities throughout the day (going to bed at night or eating at very similar times). For example, if a prediction is to be made at a time of night close to the usual bedtime, the model will tend to generate as output the activity of sleeping rather than the activity of eating. As with temperature, the hour values (hh:mm:ss) need to be transformed into values between 0 and 1. However, it is not suitable to assign an immediate numerical value to each hour-minute set, as the numerical distances would not be satisfied. For this reason, each hour-minute set is transformed into sine–cosine pairs to obtain equidistant hourly quantifications. The following expressions are used to obtain the sine and cosine of the hour-minute pairs:(7)$$\begin{array}{c} TimeX=\cos\left(\frac{2\pi \left(h+\frac{min}{60}\right)}{24}\right) \end{array}$$
(8)$$\begin{array}{c} TimeY=\sin\left(\frac{2\pi \left(h+\frac{min}{60}\right)}{24}\right) \end{array}$$
where TimeX corresponds to the cosine value of the hour and TimeY to the sine value. The parameters *h* and min of Equations (Equation 7) and (Equation 8) correspond to the hour and minute values. The result of these calculations gives a real number between −1 and 1. Therefore, as with temperature, it is advisable to normalise these values between 0 and 1.

The analysed dataset has 15 different activities labelled. However, there are records without any labelled activity in the dataset where sensor events are recorded. These records are labelled in the processing as ’Other’, thus generating activity 16. This is necessary because the model should always generate an output (or activity in this case) and, if only the 15 primary activities are taken into account, this could lead to confusion when analysing durations and repeatability. In the processing, an integer value is assigned to each activity (from 1 to 16) to obtain the output of the model.

To condition and improve the data used to train the prediction model, three stages have been designed and can be seen in Figure 3:
Sliding window method: To add more consistency to the data, a technique called ‘sliding window’ is used. The sliding window technique consists of collecting the closest old records from the sensors to obtain a certain output. That is to say, for an instant (*T*) in which an output ($Y_{T}$) is sought, the sensor registers ($X_{T-W-1}$, …, $X_{T-1}$, $X_{T}$) are used, (*W*) being the size of the window. By using the sliding window technique, greater repeatability of the data is achieved, and in addition, the network is strengthened due to the sequential pattern provided by the sensor events.Filtering repeated rows: This method serves to minimise the load of activities with longer durations. When a particular activity exceeds a certain number of consecutive records, a filtering percentage is applied to prevent the model from being trained in an unbalanced way and to avoid giving more importance to activities with longer durations. In this way, a percentage of the intermediate records of the same activity is eliminated, leaving the records at the beginning and end of the activity.Stacking and re-ordering rows: A grouping of the data windows into blocks is applied before re-ordering to maintain the sequential order of the records and then randomly re-ordering the groups of windows to avoid overfitting.

### 3.3. Neural Network Architecture

As discussed in Section 3.1.2, a neural network based on bidirectional LSTM layers was chosen to build the prediction model. A basic schematic of the information flow and the layers of the developed model is shown in Figure 4:
Split Layer: The first layer of the model is a SplitLayer, which is responsible for separating the two components corresponding to the time from the rest of the sensor information. This is due to the fact that it is more rigorous to train the LSTMs only with the information from the sensors than with all the data mixed together, as this will provide more sequential information than the time, as they are equidistant sequential patterns.Bidirectional LSTM: The bidirectional LSTM layers are the main part of the prediction model. They are responsible for providing the neural network with the sequential value corresponding to the activation of the sensors. To improve the behaviour of the neural network against overfitting and thus further generalise the model, an L2 regularisation [45] has been used. To calculate the regularisation value, it is necessary to take into account the cost function:
(9)$$\begin{array}{c} C=C_{o}+\frac{\lambda }{2n}\sum_{\omega }^{}\omega^{2} \end{array}$$The value to regularise the LSTM can be quantified using this expression Equation 9. The term $C_{o}$ refers to the original cost, while λ is the term that quantifies the regularisation weight and the proportion of $C_{o}$. The term *n* corresponds to the size of the training dataset, and ω represents all the parameters.Dropout: Dropout layers are included to reduce overfitting. They are responsible for randomly and temporarily disabling the connections between the outputs of the previous layer and the inputs of the subsequent [46] layer. Once all the data have made a pass through the model, the Dropout layers randomly change the deactivated connections while maintaining the given proportion.Batch normalisation: A batch normalisation is performed in order to reduce a term called “Internal Covariate Shift”. The internal covariate shift occurs due to the change of parameters between the layers that make up the model [47]. For this reason, the batch normalisation layer generates small batch sizes that are adjusted so that they are all approximately the same length.Concatenate layer: This layer is responsible for merging the two time components for each sliding data window back together with all the sensor information processed by the bidirectional LSTM layers to perform the final processing step.Dense layer: This is the last processing step, which takes into account both the sensor information processed by the bidirectional LSTM stage and the time of analysis. An architecture based on two fully-connected dense layers has been proposed for this system, whose distribution of neurons follows a distribution of (2/3)N and (1/3)N, where *N* is the total number of neurons.

Finally, in order to improve the understanding of the system developed in this paper, a block diagram showing all the steps to be carried out is shown in Figure 5. The figure clearly shows the two stages developed in this paper. At the top of the figure, there is the data processing stage, and at the bottom, there is the neural network stage. Once a prediction of a given activity has been made, it is checked against the data in the database. If they do not match, the system takes care of modifying the weights of the neural network layers. In addition, the main parameters of the model are included in the diagram.

## 4. Experiments and Discussion

To test the validity of the system, a supervised training of the neural network has been carried out by dividing the whole data set as follows: 70% of the total data set has been used for training, 10% for model validation and 20% for testing.

First, a summary of all the factors chosen for data processing and filtering is made. Regarding the time between samples of the Milan dataset, a total of 2 s has been chosen. For the sliding window, the 60 records prior to the prediction time (included) are taken into account. That is, the 2 min prior to the prediction time are considered. A filtering of the “Other” activity and the “Sleep” activity is carried out, taking into account only the 5% at the start and end of the activity, because these two activities are the ones with the longest duration and the most records. For the grouping of records before randomising the data, it has been decided to group them into pairs of two windows in order to maintain the temporal sequence of events.

The hyperparameters selected to model the neural network are detailed below. For the bidirectional LSTM layers, 64 cells have been used for the output layer, and a very small value for the L2 regulariser ($1\times 10^{-6}$), since it is advisable to use a value close to zero for the regulariser to have beneficial effects for the model. This is because if a high value is chosen, the method attacks the model so much that it loses convergence. The dropout value at the output of the bidirectional LSTMs is 0.2, and the dropout at the end of the network is 0.4. This dropout distribution has been selected because it is advisable to deactivate the links between neurons progressively. That is to say, for layers closer to the beginning of the model, it is preferable to apply little dropout, and for layers closer to the end of the model, it is gradually increased until the difference between training and validation is very small. With respect to the number of neurons, a total of 6600 neurons has been established: 4400 neurons for the first dense layer and 2200 neurons for the second dense layer. A distribution of (2/3)N and (1/3)N has been followed because, although it causes the neural network model to train more slowly, it offers better results in terms of accuracy for more advanced epochs.

Finally, it is necessary to include general training factors. A total of 140 epochs were run with a batch size of 256. The model took 630 min to train on an Intel(R) Core(TM) i9-10900K CPU@3.70 GHz/128 Gb with two RTX3090 GPUs.

In Table 3, the accuracy, recall and F1-score values can be checked. The final result of the test is a value of 0.9542 for the accuracy and a value of 0.184 for the loss. The graphs of the evolution of the accuracy and loss as the epochs progress during training are also included, both for the training dataset and for the validation dataset (see Figure 6):

It can be seen that both training and validation stabilise around epoch 140. It can also be seen in the figure that there is hardly any overfitting, as the training curve and the validation curve are very close to each other. This is beneficial for the model, as it will fit well to unknown data that have not been used during training.

To visualise the dispersion that exists between the predictions, a confusion matrix generated as a result of the application of the test is included below (see Table 4):

It can be seen that the most influential weights are distributed along the diagonal of the confusion matrix. This means that most of the predictions are accurate. A positive aspect that can be extracted from the results is that the prediction model correctly differentiates activities that occur in the same room or that are very close in space, as can be the case of Sleep and Master Bedroom Activity. Both activities are in the same room and share the same sensors; however, the network is able to correctly differentiate which one is being carried out thanks to the time of day or the habit of the resident. Another positive aspect is the ability to correctly differentiate the Eve Meds activity from Morning Meds. By looking at both activities, it can be seen that the process of performing the activity is identical. However, because the model uses the time of day, the two activities can be classified separately. This is very important when monitoring medication schedules.

Furthermore, thanks to the confusion matrix, it is possible to see what the most common failures are, and the following conclusions can be drawn:Failure to detect taking medication: This failure occurs in both the Eve
Meds activity and the Morning
Meds activity, and both are confused with Kitchen
Activity. This is because the box where the medicines are stored is in the kitchen, and although this box has a contact sensor, there may have been some malpractice during the day, such as leaving medicines out of the box or leaving the door open.Confusion between Sleep and Bed
to
Toilet: This problem can occur because the two rooms are in close proximity and can create confusion for motion sensors. Furthermore, the Bed
to
Toilet activity is short-lived and not as robust as the Sleep activity (which has a duration of many hours).Confusion between Dining
Room
Activity and Read: The Read activity takes place exclusively in that room. Therefore, it is an activity included within the range offered by Dining
Room
Activity.Tendency to confusion with Other: The Other activity is the dummy activity generated to label times when the user does nothing. For this reason, the neural network may fail during the transition between periods of time with no activity and a new activity. Specifically, the highest failure rate in this respect falls on the Sleep activity, as it is the activity that most closely resembles Other in terms of duration.

These failures could be reduced by balancing all the activities so that they all have the same number of occurrences.

In a first step, the training of the system and its subsequent predictions were carried out using an architecture based on simple unidirectional LSTM layers. In other words, a vLSTM architecture was used. However, the results obtained with this architecture were not very high, giving a result of 85% in the best test performed. Thus, the work was done to reinforce the information of the subsequent events by converting the vLSTM architecture to one based on bidirectional LSTM layers.

In order to obtain such high accuracy and reliability results, many network tuning tests have been carried out by varying the hyperparameters gradually. To achieve this goal, a range of different percentages for the dropout layers has been tested until the overfitting has been reduced as much as possible. For example, for the dropout layer at the model output, a value of 0.2 generates a very large and unacceptable overfitting. However, a value of 0.6 provides very similar overfitting to that obtained with a value of 0.4 but with much lower total accuracy and F1-score values. For this reason, it was decided that a value of 0.4 was optimal. Different sizes for the sliding window were tested until a balance was reached between the success of activities with longer durations and activities with shorter durations. The optimal window value was selected to be 60 samples, as initially, 30 samples in total were tested (in order to carry out tests of 1 min duration). However, with such a small number of samples, the system was worse at recognising activities of long duration. Similarly, testing with a window size of 120 samples (4 min of duration), the system was worse at recognising short duration activities. For this reason, it was decided to set a window size of 60 samples to obtain a balance in the recognition of activities with such variable durations. Different numbers of cells have been used for the bidirectional LSTM layers with different values for the regularisers until a high hit rate was achieved. A value of $1\times 10^{-6}$ was chosen for the L2 regulariser because, using a value of $1\times 10^{-5}$, the accuracy and F1-score of the system would drop exponentially. For this reason, it is preferable to use a very low value for the L2 regulariser so that it affects the overfitting as much as necessary without greatly reducing the overall results. Finally, it is also necessary to add that a balance has been reached in the number of neurons for the dense layers in order to obtain a model that is not very slow computationally and that achieves high results. It was attempted to increase the number of cells in the bidirectional LSTM layers from 64 to 128, but this increased the computational burden so much that the test was deemed unnecessary due to the small improvement achieved. The same behaviour occurred when increasing the number of neurons in the dense layers.

It is important to note that, before obtaining the final model proposed in this work, the development of a model whose data processing was based on the use of data windows with a fixed number of events (regardless of time) without temporal disaggregation was initiated. This resulted in very low and highly variable accuracies depending on the activity analysed. This approach also allowed real-time prediction, which is one of the main objectives of this research, but with very poor results.

There are other more conventional methods that also aim to recognise activities in a domestic environment. For example, the work developed by D. Singh et al. [48] focuses on the detection of activities in three different houses through the use of recurrent neural networks. Although the dataset is different from the one used in the present work, a lower success rate can be observed than the one achieved with bidirectional LSTM networks [49]. Another interesting work to compare with is the one developed by D. Anguita et al. [50] due to the use of SVM for prediction. The authors reach an accuracy of 89.3% in the detection of six activities by analysing the information obtained by the use of a smartphone. Although the SVM method is quite reliable, it is computationally expensive, which benefits the use of neural networks to minimise the time load. Finally, another work to be taken into account is the one developed by R. Saeedi et al. [51], as they develop a very complete neural network architecture to recognise activities using wearable sensors. This architecture is based on the use of convolutional and max pooling layers to end up with two unidirectional LSTM layers. In total, this architecture has 11 layers and achieves an F1-score of 70%. Although working with different data, it is understood that the use of an architecture based on bidirectional LSTM layers and dense layers improves the performance of the system, as the work developed in this paper achieves an F1-score of 95%.

In order to check the reliability and accuracy of the proposed model, a comparison has been made with the work carried out by D. Liciotti et al. [36]. These authors developed a model based on a variable data window, which stores the total number of events corresponding to a given activity. This means that each activity presents a number of events of variable size, these events being the ones corresponding to the totality of the activity and not to the transitions. Due to this reason, the training of the model is performed knowing the duration of the activity, a reason that is not beneficial when making predictions in real-time, since in this situation, the duration of the activity being performed is unknown. This is why, in this paper, a model based on fixed sampling has been proposed in order to make real-time predictions at equispaced times. With respect to the percentage of success, the present paper obtains a result of 95.42%, while the system developed in the paper by D. Liciotti et al. achieves a result of 94.12% for the same database and the same type of neural network. This represents an improvement of 1.3% with respect to the existing models. Finally, it is necessary to note that these authors did not contemplate real-time implementation, seeking only a prediction on training, validation and test data. This disadvantage has been solved with the system proposed in this paper.

A summary of the comparisons that have been made between the model developed in this work and the contributions of other authors can be found in Table 5. In this table, it is possible to observe in detail the tools and datasets that other authors have used to perform HAR, the success rate that they have achieved and the total activities that they are able to recognise. As can be deduced, it is very difficult to make a comparison of results with other authors who use other datasets, other technologies and other methods to approach HAR. We are interested in the recognition of a large number of activities of older people in their homes while respecting their privacy to make recommendations in real-time when deviations from established patterns or risk situations occur. For this reason, we have chosen to make comparisons with the work carried out by D. Liciotti et al. [36] as they use a similar method to the present work, and they have chosen the same dataset. The Milan dataset has been chosen in the present work because it presents a large number of data and deals with the most common activities at home.

## 5. Conclusions

This paper presents an effective prediction method to identify in real-time the most common daily activities performed by persons living alone during their daily routine. The solution is capable of detecting a wide range of activities, some of them being critical to health, such as taking medication or eating the right meals for the day. This improves the independence of elderly people, giving them greater autonomy and improving their quality of life.

The proposed prediction model is composed of a recurrent neural network, which is fed with a set of pre-processed and filtered data. The database chosen to train the prediction model contains records from 33 simple and unobtrusive sensors that are installed in a house where a single person resides with her pet for approximately 84 days. These data go through a treatment and filtering process composed of three stages: a sliding window method, filtering repeated rows and stacking and re-ordering rows. This treatment process performs a fixed-time sampling along the dataset, allowing a prediction model to be generated that can be used in real-time, since it is only necessary to make requests to the sensors to know their status at equispaced times.

Comparing the model proposed in this paper with similar works by other authors, an improvement in functionality can be seen. The model is able to perform real-time prediction once the training stage has been completed, giving the system a functional capability. In addition, the percentage of accuracy achieved by the prediction model is higher than that of other works taken as a reference.

The system proposed can be very useful when supervising the tasks performed by a dependent person at home. This system is a very important support tool for the specialists in charge of this function, as it allows the immediate detection of alterations in the behavioural patterns of the subject. This is a breakthrough that can enable residents of a specialised care centre to remain at home for longer periods of time even when they are alone, allowing therapists to monitor their daily activities remotely in order to watch over their health.

Thus the model presented in this paper is based on data generated by a network of sensors to predict the activity that a person is doing alone. Future work is proposed to generalise the model so that it can be effective in homes where several users live. Similarly, the implementation of a hidden Markov model that reinforces the output of the network with the information obtained from the daily habits of the user is also proposed in order to avoid the accuracy failures detailed in the experiments.

## Figures and Tables

**Figure 1 sensors-21-05270-f001:**
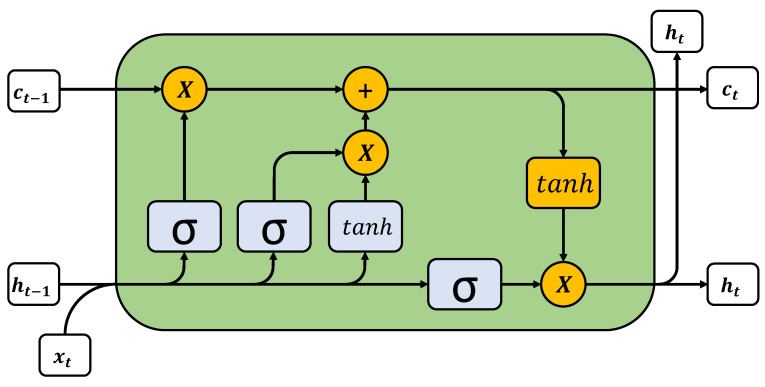
Components of an LSTM cell.

**Figure 2 sensors-21-05270-f002:**
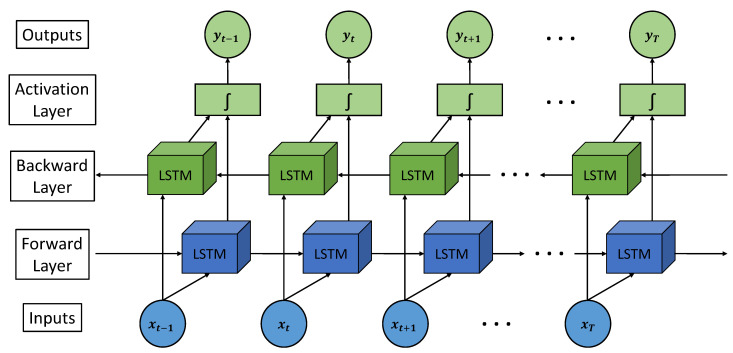
Architecture of a bidirectional LSTM network.

**Figure 3 sensors-21-05270-f003:**
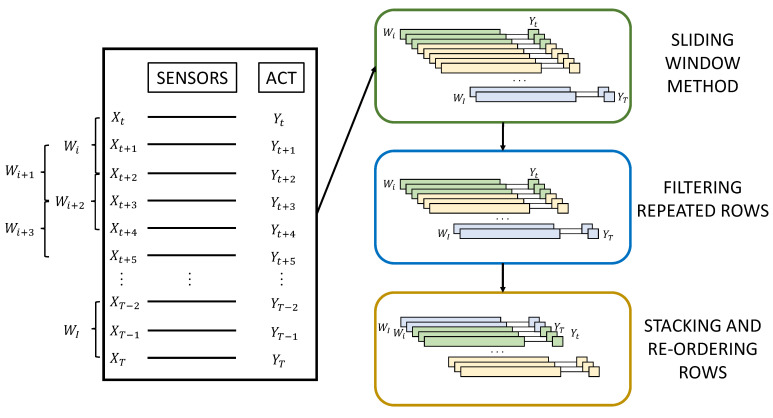
Processing methods and treatment of training data.

**Figure 4 sensors-21-05270-f004:**
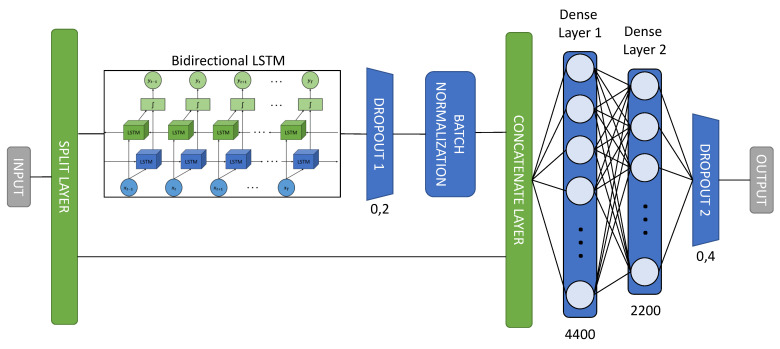
Neural network architecture.

**Figure 5 sensors-21-05270-f005:**
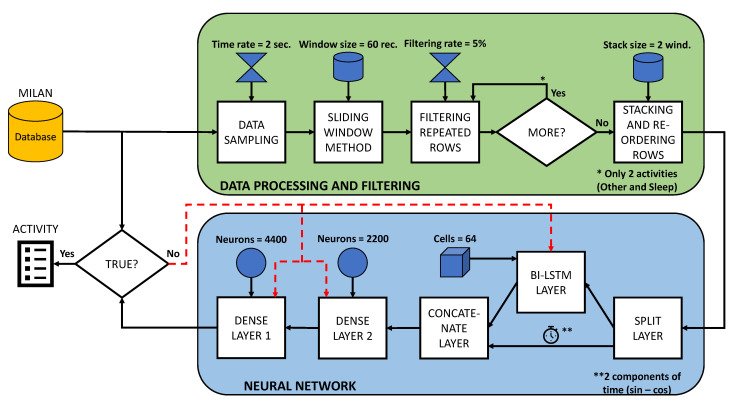
Block diagram of the system developed.

**Figure 6 sensors-21-05270-f006:**
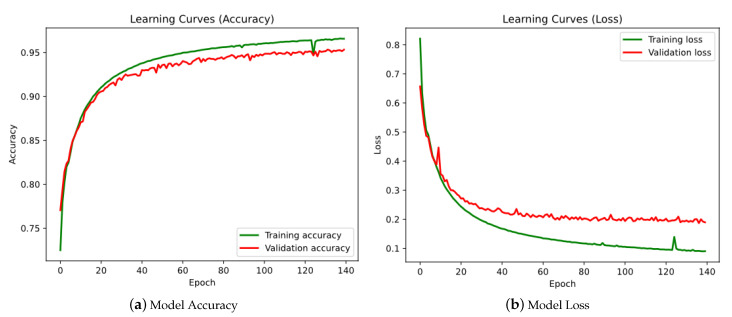
Bi-directional LSTM-based neural network learning curves.

**Table 1 sensors-21-05270-t001:** Milan dataset main features.

Type of Sensors	Time Range	Activities	Activities (Instances)
Motion (M) Door (D) Temperature (T)	16 October 2009– 6 January 2010	15	Bed to Toilet (89), Desk Activity (54), Chores (23), Dining Room Activity (22), Evening Medication (19), Read (314), Kitchen Activity (554), Guest Bathroom (330), Leave Home (214), Master Bathroom (306), Sleep (96), Master Bedroom Activity (117), Meditate (17), Morning Medication (41) and Watch TV (114)

**Table 2 sensors-21-05270-t002:** Milan dataset example.

Timestamp $t_{i}$	Sensor	State	Activity
2009-11-19 08:47:38.000019	M017	ON	Guest Bathroom begin
2009-11-19 08:47:38.000021	M015	OFF	
2009-11-19 08:47:40.000041	M011	OFF	
2009-11-19 08:47:40.000089	M018	ON	
.	.	.	
.	.	.	
.	.	.	
2009-11-19 08:49:02.000086	M018	ON	
2009-11-19 08:49:08.000076	M017	ON	Guest Bathroom end

**Table 3 sensors-21-05270-t003:** Summary table of results by activity.

Activity	Precision	Recall	F1-Score
Bed to Toilet	0.93	0.92	0.93
Chores	0.95	0.97	0.96
Desk Activity	0.98	0.98	0.98
Dining Rm Activity	0.97	0.96	0.96
Eve Meds	0.90	0.94	0.92
Guest Bathroom	0.93	0.97	0.95
Kitchen Activity	0.97	0.97	0.97
Leave Home	0.93	0.96	0.94
Master Bathroom	0.96	0.95	0.96
Meditate	0.95	0.99	0.97
Watch TV	0.97	0.97	0.97
Sleep	0.90	0.93	0.91
Read	0.97	0.97	0.97
Morning Meds	0.92	0.91	0.91
Master Bedroom Activity	0.96	0.94	0.95
Other	0.90	0.88	0.89
Accuracy			0.95
Macro avg	0.94	0.95	0.95
Weighted avg	0.95	0.95	0.95

**Table 4 sensors-21-05270-t004:** Neural network confusion matrix.

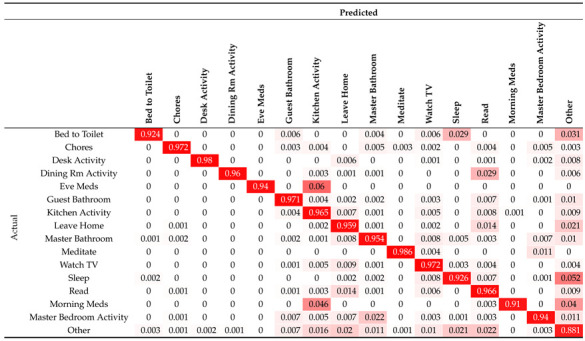

**Table 5 sensors-21-05270-t005:** Comparision with other similar works.

Paper	Dataset	Success Rate	Activities	Sensors	DL Method
[48]	Own development [49]	89.80%	10	Non-intrusive	Recurrent neural network
		85.70%	13		
		64.22%	16		
[50]	Own development [50]	89.30%	6	Smartphone	SVM algorithm
[51]	OPPORTUNITY	70.00%	17	Wearables	CNN-LSTM neural network
[32]	UCI [34]	95.09%	6	Smartphone	iSPLInception
	OPPORTUNITY	88.14%	17	Wearables	
	Daphnet	93.52%	2	Wearables	
	PAMAP2	89.09%	11	Wearables	
[33]	UCI [34]	98.85%	6	Smartphone	DL-MFAP
[35]	UCI [34]	88.14%	6	Smartphone	Deep LSTM Network
[36]	Milan (CASAS)	94.12%	16	Non-intrusive	Bidirectional LSTM
Proposed	Milan (CASAS)	95.42%	16	Non-intrusive	Proposed architecture

## Data Availability

Publicly available datasets were analyzed in this study. This data can be found here: http://casas.wsu.edu/datasets/ (accessed on 29 June 2021).

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
