# Peer review of "Daily Human Activity Recognition Using Non-Intrusive Sensors"

_sensors, 2021, doi:10.3390/s21165270_

Round 1

Reviewer 1 Report

The paper suggests a useful forecast technique with the aim of recognizing everyday activities done by persons living alone.

The English of the paper should be improved e.g.:

if she/he has follow healthy habits -> if she/he has followed healthy habits

presents a concept of internal memory unit which very beneficial -> presents a concept of internal memory unit which is very beneficial

Another correction is the name of the method that was mentioned in section 2. The authors called it "Naive Bayer", but the correct name is "Naive Bayes".

One more thing - The authors write "Monitoring the activities performed by an elderly person can inform, in real time, of such undesired behaviour as falls". This is obviously correct, but nowadays the real-time alerts are also for equipment, not just for persons as was written in Y. Wiseman, "Take a Picture of Your Tire!", Proc. IEEE Conference on Vehicular Electronics and Safety (IEEE ICVES-2010) Qingdao, ShanDong, China, pp. 151-156, 2010. I would encourage the authors to cite this paper and mention that sensors can also alert about equipment failures which can also improve safety.

Author Response

Dear reviewer,

Best regards,

The authors.

Reviewer 2 Report

Dear Authors, 

Please find the attached file for my comments.

Best Regards

Reviewer

Author Response

(The authors gave the same response as above.)

Round 2

Reviewer 1 Report

The paper in its current form is certainly publishable, so I recommend accepting the paper.

Reviewer 2 Report

Dear Authors,

Thank you for addressing all my comments and the paper is accepted from my side. 

Best Regards